# Quality of Life and Related Factors in Patients Undergoing Renal Replacement Therapy at the Hospital General Universitario de Ciudad Real: Cross Sectional Descriptive Observational Study

**DOI:** 10.3390/jcm12062250

**Published:** 2023-03-14

**Authors:** Jose Miguel Berenguer-Martínez, Rubén Jose Bernal-Celestino, Antonio Alberto León-Martín, María Teresa Rodríguez González-Moro, Nuria Fernández-Calvo, Leticia Arias-del-Campo, Margarita Civera-Miguel

**Affiliations:** 1Department of Nursing, Universidad Católica de Murcia (UCAM), 30107 Murcia, Spain; 2Research, Teaching and Training Department, University General Hospital, Castilla-La Mancha Health Service (SESCAM), 13005 Ciudad Real, Spain; 3Department of Nursing, Ciudad Real Nursing Faculty, University of Castilla-La Mancha (UCLM), 13071 Ciudad Real, Spain; 4Quality Department, University General Hospital, Castilla-La Mancha Health Service (SESCAM), 13005 Ciudad Real, Spain; 5Department of Medical Sciences, Faculty of Medicine, Ciudad Real, University of Castilla-La Mancha (UCLM), 13071 Ciudad Real, Spain; 6Department of Nefrology, University General Hospital, Castilla-La Mancha Health Service (SESCAM), 13005 Ciudad Real, Spain

**Keywords:** quality of life, hemodialysis, peritoneal dialysis, kidney failure, chronic disease progression

## Abstract

**Background:** The aim of the present study was to determine the relationship between the quality of life of patients on renal replacement therapy and the Symptomatology they presented. **Methods:** Cross-sectional descriptive observational study: quality of life was assessed by means of the KDQOL-SF questionnaire, Symptomatology by the Palliative Care Outcome Scale-Symptoms Renal questionnaire, and sociodemographic and clinical data of patients in the Hemodialysis Unit (HD) of the Hospital General Universitario de Ciudad Real (HGUCR) by means of personal interviews and clinical history data. **Results**: A total of 105 patients participated in the study, 63 (60.57%) men and 42 (40.38%) female. The mean age was 62.5 dt (14.84) years. Of these, 43 (41%) were on peritoneal dialysis and 62 (59%) were on hemodialysis. The mean quality of life score was 44.89 dt (9.73). People on hemodialysis treatment presented a better quality of life than those on PD treatment: 49.66 dt (9.73) vs. 38.13 dt (9.12) t = 7.302, *p* < 0.001. A higher score on the symptom impairment scale (post-renal) correlated with worse scores on the total quality of life score: r = −0.807, *p* < 0.001. It was observed that those who improved the distress symptom scored better on the total quality of life questionnaire: 50.22 dt (8.44) vs. 46.42 dt (9.05), *p* < 0.001. **Conclusions**: The presence and management of the large number of symptoms that appear as side effects, such as distress or depression, could determine changes in some components of quality of life.

## 1. Introduction

Chronic kidney disease (CKD) has been recognized as a growing global public health problem; an estimated 1.2 million people died in 2015 from kidney disease [1]. The prevalence of having the disease is estimated at 13.4%, and patients with kidney disease who will need renal replacement therapy (RRT) is estimated to be between 4902 and 7084 million people [2]. CKD is closely related to cardiovascular risk [3], in addition to other risks treated and observed through the implementation of the KDQOL-SF, quality questionnaire in the unit, such as psychological, depression, and lack of energy, where we can observe decreased risk of quality of life in people undergoing hemodialysis and peritoneal dialysis treatment [4], despite knowing specific interventions such as hiking that favor not having depression through physical exercise [5]. The global increase in this disease is mainly due to the increase in the prevalence of risk factors associated with the disease, where we can mainly observe diabetes mellitus, hypertension, obesity, and increased longevity in the population [6]. Treatments based on a balanced diet are already being used to improve the living conditions of the population and reduce risk factors based on dietary interventions and exercise [7]. The government of Spain in 2020 affirmed that 49.9 million donors were available to perform organ explants, of which 3423 kidney, 1227 liver, 300 cardiac, 419 pulmonary, 76 pancreatic, and 4 intestinal transplants were performed, as stated by the National Transplant Organization (CNT) as of 31 December 2019 [8]. Despite the research work and activity in the surgical units, the reality is that a significant number of patients persist on the waiting list, pending an organ, in addition to how the SARS-CoV-2 pandemic has affected this time [9]. In 2022, the CNT again presented the results of the transplants carried out, reaching historical levels in the year 2020, when 3402 kidney transplants were performed (15% more than the previous year), increasing lung transplants by 15%, pancreas by 12%, cardiac by 3%, and intestinal by 4%. Living-donor kidney transplantation activity this year represented 10% of all transplants [10]. The need to offer scientific societies added value through studies vindicates the justification that transplant patients have a better quality of life than patients in treatment, although certain factors such as socioeconomic factors, age, level of education, and population influence [11]. We believe it is necessary to routinely implement the assessment of quality of life. Our study has evaluated in a single instance, through a question, if there has been a failed kidney transplant before, of which 20% of the subjects answered that they had received a failed transplant, and 80% had not received a transplant and were on active dialysis treatment, which shows that people with failed kidney transplantation have a poor quality of life [12]. Of patients, 80%, bearing in mind their age profile and associated pathologies such as hypertension, hypercholesterolemia, and diabetes, might have difficulty accessing the option of kidney transplantation, in addition to the age factor that is decisive since it limits the survival of the donated graft [13]. The results of this present study establish a comparison of the quality of life in patients on hemodialysis and peritoneal dialysis; it is true that the quality of life of transplant patients increases as long as the person is more independent for activities of daily living, in addition to not undergoing dialysis treatment in any of the modalities [14]. At the same time, we must take into account the work of nursing professionals in both types of patients in educational and psychological care areas in order to improve the process before kidney transplantation [15]. In addition to offering psychological support by specialized personnel to face the disease [16].

Knowing the quality of life of patients with ACKD and the related factors that affect it the most would lead to a scenario for proposing hygienic dietary measures, lifestyle changes, and educational interventions, which have already been shown to be effective, such as physical exercise. Furthermore, assessing HRQoL in patients with kidney disease could be an important tool for healthcare professionals and become an integral part of patient-centered care [17].

Understanding quality of life is important for improving symptom relief, patient care, and rehabilitation. Problems revealed by patients’ self-reported quality of life may lead to modifications and improvements in treatment and care or may show that some therapies offer little benefit. This type of information can also be used to help future patients anticipate and understand the consequences of their disease and its treatment [18].

The aim of the present study is to determine the relationship between the quality of life of patients with ACKD on RRT and their presenting Symptomatology.

## 2. Materials and Methods

### 2.1. Design

An observational, transversal, and descriptive study was carried out as part of the “Project presented at the General Hospital of Ciudad Real” to know the quality of life of patients in renal replacement therapy.

### 2.2. Scope and Participants

Patients in stage IV RRT are in the Hemodialysis Unit (HD) of the General University Hospital of Ciudad Real (HGUCR). Chronic patients with active treatment are in the indicated period: 2020–2021. Acute patients requiring dialysis due to clinical urgency or children, patients with cognitive deficits, and patients with language barriers were excluded.

A total of 105 patients were selected from the unit who knew the selection criteria and wished to participate in the study.

Data collection was conducted from 1 November 2020 to November 2021. Participation was offered to patients attending the hemodialysis service for renal replacement therapy (hemodialysis or peritoneal dialysis). Once the informed consent form had been signed, the variables extracted from the clinical history were collected. At the same time, a personal interview was conducted to collect the variables related to quality of life and symptoms that were not reflected in the clinical history.

The variables collected in the present study can be classified as the following sociodemographic characteristics: sex, age, highest level of education, occupation, marital status, and type of cohabitation (personal interview).

### 2.3. Clinical Variables

The etiology of ACKD, time on HD (months), previous failed renal transplant, hematocrit, hemoglobin, EPO administration (%), total protein (g/dL), albumin (g/dL) of the last analysis, and Kt/V (dose) were all clinical variables.

Symptoms in advanced chronic kidney disease will be obtained by administering the Palliative Care Outcome Scale-Symptoms Renal (POS-S Renal) questionnaire [17,18]. This instrument was developed by the King’s College London team and validated in Spanish by Gutierrez et al. [19]. The POS-S Renal is an instrument for the evaluation of symptoms that has demonstrated its usefulness in the evaluation of symptoms in patients with ACKD. It is an easy, short self-questionnaire that was developed to evaluate symptoms in ACKD. It consists of 17 symptoms with a 5-point Likert-type response format, where the patient assigns how each symptom has affected them to a category ranging from “none” (0) to “unbearable” (4), increasing in intensity with higher scores. This tool asks the patient about symptoms perceived during the last week and also allows the addition of other symptoms that do not appear in the questionnaire. It evaluates the presence of pain, shortness of breath, weakness or lack of energy, nausea (as if feeling like vomiting), vomiting, poor appetite, constipation, mouth problems, drowsiness, poor mobility, itching, trouble sleeping, restless legs or difficulty keeping legs still, feeling anxious and/or depressed, skin changes, diarrhoea, cramps, and other symptoms, as well as the symptom that has affected you the most and the one that has improved the most.

Quality of life variables are as follows: physical, psychological, and social functions; general perception of health, mobility, and emotional well-being; among others, were evaluated by means of the K Questionnaire.

### 2.4. Data Analysis

#### 2.4.1. Descriptive Statistics

Qualitative variables were presented with their frequency distributions; for quantitative variables, the mean and standard deviation were determined.

#### 2.4.2. Inferential Statistics

To evaluate the relationship between the different variables and quality of life, Pearson’s correlation was used for quantitative variables, and a Student’s *t*-test for independent samples was used for qualitative variables. For the study of quantitative variables, a prior analysis of normality was performed. Considering the estimated sample size, the Kolmogorov-Smirnov test was used in this study. Non-parametric tests were used as an alternative when parametric tests could not be carried out. A confidence level of 95% was used for all calculations. The protocol for this study was approved by the Research and Ethics Committee of the Hospital General Universitario de Ciudad Real.

## 3. Results

Finally, 105 patients participated in the study, of whom 63 (60.57%) were men and 42 (40.38%) were women. The mean age was 62.5 dt (14.84) years. Of these, 43 (41%) were on PD treatment, while 62 (59%) were undergoing HD sessions. The mean time on HD treatment was 53.33 dt (73.49) months and on peritoneal dialysis, 17.29 dt (15.70) months.

Some analytical parameters are presented in Table 1.

### 3.1. Symptoms

The most common symptoms presented by the patients were headache, 90 (85.7%), asthenia, 83 (79.05%) and weight loss, 53 (50.48%). It was found that fever, pyuria, lymphadenopathy, and hematuria were more frequent in people treated with peritoneal dialysis, while asthenia, weight loss, and headache were more common in people on hemodialysis (Table 2).

In terms of symptom involvement during the last week, participants reported that sleeping problems, weakness or lack of energy, and cramps were the symptoms that most affected their daily lives. In contrast, diarrhoea, shortness of breath, and poor appetite were the least bothersome symptoms (Table 2). The mean score of the renal POS was 23.95 dt (11.35), measured as a Likert scale, with a higher score obtained by those on peritoneal dialysis, 34.71 dt (10.00), compared to those treated with hemodialysis, 17.80 dt (6.44), *t* = 10.072, *p* < 0.001 (Table 3).

To the question, “Which of the above symptoms had affected you the most?”, the patients answered a lack of energy, 43 (41%). The symptom that had improved the most in the last week was cramps, 26 (24.8%) (Figure 1).

### 3.2. Quality of Life

The majority of participants, 53 (51.1%) defined their health status as “bad” or “very bad”.

In terms of health-related quality of life, the mean score of the KDQOL-SF questionnaire was 44.89 dt (9.73). The majority of the participants, 75 (72.1%), CI 95% of (63.35 to 80.88) had a low quality of life (less than 50 points on the KDQOL-SF). Age was correlated with the Kidney Disease Effect score: r = −0.377, *p* < 0.001 and negatively with the SF-12 Physical Component: r = −0.401, *p* < 0.001. Likewise, time spent on PD was related to a lower score in the Symptoms or Trouble List dimensions: r = −0.362, *p* < 0.001 and Effect of Kidney Disease: r = −0.355, *p* < 0.001, and Total Score: r = −0.346, *p* < 0.001.

We found no relationship between any component of quality of life and the analytical parameters. The statistical test for Quality of Life Spearman’s Rho (Rho) was used to analyze the quality of life. The results from the statistical tests are provided. People on hemodialysis treatment presented a better quality of life than those on PD treatment on the total KDQOL-SF scale. In addition, people on HD showed higher scores in the domains of Symptoms or Problem List, Effect of Renal Disease, and Mental Component compared to those on PD (Table 4).

Higher symptom affect scale score (POS-S Renal) correlated with worse scores on the Total Quality of Life Score: r = −0.807, *p* < 0.001; Symptomatology or Problem List: r = −0.880, *p* < 0.001, Disease Effect: r = −0.688, *p* < 0.001; Disease Burden: r = −0.407, *p* < 0.001; and Mental Component: r = 0.058, *p* < 0.001. However, this scale score did not correlate with the Physical Component of the SF-12: r = 0.049, *p* = 0.638. Those who had presented improvement in symptoms such as cramps and distress had a better score in the Kidney Disease Effect domain.

Patients who indicated that they had improved symptoms of depression and distress had better scores on the Symptoms or Problem List dimension than those who had not improved those symptoms. It was observed that those who improved the distress symptom scored better on the total Quality of Life questionnaire. (Table 5).

Similarly, it was observed that people who had improved symptoms, such as itching, scored lower on the Symptoms or Problem List, Illness Effect, Mental Component, and Total Scale Score components compared to those who did not indicate such improvement. Patients who reported improvement in the constipation symptom had lower scores on the Symptoms or Problem List component compared to those who did not indicate such improvement (Table 5).

In addition, Table 6 shows the results of the multivariate analyses. Only those variables that were significant in the bivariate or multivariate analysis were included.

## 4. Discussion

This study found that the main symptoms presented by people receiving renal replacement therapy are affected in the course of treatment, with increased associated symptoms. The main symptoms we observed in patients on peritoneal dialysis were as follows: fever, pyuria, and hematuria. However, patients on hemodialysis with the most frequent symptoms were as follows: asthenia, weight loss, and headache. Both groups (hemodialysis and peritoneal dialysis) reported that sleep problems, with the consequent need for medication; weakness, lack of energy, and cramps were the symptoms that affected their day-to-day. Patients treated with PD obtained higher scores on the Palliative Care Outcome Scale-Symptoms Renal scale, which, like other studies, reveals that patients evaluated in RRT have higher scores with similar symptoms [17]. In this study, there has not been a significant mortality rate in comparison to other studies of a longer duration, given that none of the 105 patients were lost over the course of the year. With regard to quality of life, among participants who completed the KDQOL-SF and PoS-S Renal questionnaires, the dimension of physical functioning was correlated with younger age and the start date of treatment. However, patients with more treatment time correlated with a higher risk of suffering psychological alterations such as depression, insecurity, and in some cases even psychiatric pathology, similar to the findings of other studies where the time in renal replacement therapy has a not only physical component but also a mental component, and specific interventions based on physical exercise are already performed to promote mood [5]. The most affected areas of both groups related to quality of life were the physical component and burden of the disease, and the least affected was the dimension of Symptomatology. Perhaps the psychological factor is an essential component to maintaining health in chronic pathologies, to be able to establish interventions and reinforce this aspect as happens in other institutions, where they reveal however that the Burden of the Disease and Symptomatology are what most affect patients with CKD [18]. As for the experience, patients who receive non-failed renal treatment have greater vitality and motivation to live.

The most affected areas when analyzing the quality of life through the KDQOL-SF, questionnaire were the dimensions of the physical component and Burden of the Disease while the one that offered the lowest score was the dimension of Symptomatology. Age correlated with the Effects of the Disease dimension, so that the older people obtained better scores in this section. It could be due to the fact that with age, there is a greater adaptation to its effects. However, older people had worse scores in the SF-12 Physical Component dimension, in line with the progressive deterioration that they develop with the progression of kidney disease. In general, people who had been treated for PD longer had lower scores in the sections of Symptoms, Effect of the disease, and Total score of quality of life. People on hemodialysis had a better quality of life than those on PD treatment. In addition, people on HD showed higher scores in the domains of Symptoms, Effect of Kidney Disease, and Mental Component than those who underwent PD. The involvement of symptoms (POS-S Renal) was correlated with worse total quality of life and worse scores in the sections Symptomatology or List of Problems, Effect of the disease, Burden of the disease, and Mental Component SF-12. It can be observed that people who reported improvement of certain symptoms in the last week, such as distress, obtained a higher score in the quality of life category in general, Effect of the Disease, and Symptoms. As mentioned above, our study showed a higher quality of life in different sections of people with HD versus PD. In line with our results, Gonçalves’ study concluded that, although peritoneal dialysis obtained higher scores in three domains of KDQOL-SF, compared to two in hemodialysis, patients with the latter modality showed a better quality of life because the domains in which hemodialysis is better have a greater influence on well-being [2]. In our study, we only found better quality of life in PD patients in the Burden of the Disease section, although the differences were not statistically significant. Our results are consistent with one review, in which we only found a difference in quality of life in three studies based on the type of therapy they received. In two of them, the quality of life was higher in people treated with hemodialysis, and only one found a difference in favor of peritoneal dialysis in the section on Effect of Kidney Disease [20].

Other studies have reported equivalent quality of life for most domains in HD and PD patients, although there were differences in satisfaction with care that favored PD while satisfaction with physical health and symptoms of depression favored HD [13]. Other studies highlight the importance of certain clinical factors (such as anemia and bone mineral disease) and sociodemographic factors (such as employment problems and poverty) that dialysis patients have to constantly deal with more than the treatment itself [1]. However, a review with a more recent meta-analysis found that patients with PD had a higher HRQoL than patients with HD in the subdomains of physical functioning, role limitations due to emotional problems, and the effects and burden of kidney disease [12].

In any case, it is necessary to know that there are aspects of quality of life that are important for the patient and that are not included in conventional scales. In addition, these aspects do not have to be common to all patients or defined by the type of treatment applied to them. Thus, each patient has a different concept of each component of quality of life. For example, having physical well-being could relate to being able to do things independently, having control over symptoms, maintaining physical health, and being alive. Having social support could be considered in several aspects: having practical social support, social-emotional support, socialization, etc. In fact, studies showed that perspectives on the meaning of quality of life in each field could vary according to the state of frailty. For example, being alive meant day-to-day survival for frail participants but included the desire for new life experiences for non-frail participants [21]. Limitations of the study included a lack of clinical data such as laboratory tests, ultrafiltered and Kt/V, or social data that could interfere with the patient’s quality of life. Likewise, the cross-sectional nature of our study does not allow us to draw conclusions about the causal or temporal relationships between the dialysis modality, the presence of symptoms, their improvement, and Quality of Life. Nor is it possible to know if there could be variables prior to the beginning of therapy that influenced the quality of life and that conditioned the treatment applied. We consider the need for further intervention-based studies to improve symptoms that may affect quality of life. Longitudinal follow-up studies are also necessary to evaluate the quality of life before the beginning of the proposed therapy and after it in order to know exactly how they influence the patient’s day to day. We consider it necessary to apply validated instruments and scales, such as KDQOL-SF, and renal POS-S, to know the current situation of the disease as well as the main symptoms presented by patients in dialysis units. Patient health education could help address current and future issues, improve communication with healthcare staff when making shared decisions, and improve their quality of life. The quality of life in patients undergoing dialysis treatment, regardless of the treatment modality, is essential to knowing the effectiveness of adherence to treatment. The result of several studies is based on quantifying the characteristics of water restriction, medications, and dialysis treatment with different dialyzers since these are associated with an increased risk of mortality and adverse patient outcomes. Therefore, patient education and motivation are essential to improving and reducing water gain before treatment, whether in hemodialysis or peritoneal dialysis [19]. Water gain, as well as increased flow from dialysis sessions, is associated with hypertension, left ventricular enlargement, and consequent cardiovascular risk [22]. However, other studies have focused on knowing the neurological status of the patient through blood oxygenation in patients with arteriovenous fistula (AVF) with low flows by analyzing the oxygenation coefficients and flow velocity using the effective coefficient (COef) [23]. The main markers that we observe in the scientific literature are based on the treatment modality, glomerular filtration rate, use of one or another dialyzer, dialyzer membranes, nutritional status, adherence to treatment, as well as clinical values of creatinine, urea, albumin, sodium, potassium, calcium, mainly in patients on renal replacement therapy (RRT), as well as body composition by bioimpedance analysis to know this clinical status, musculoskeletal, and hydric patient prior to the start of RRT [24]. It is difficult to find trials of a similar nature offering specific interventions to provide similar data due to the very diverse nature of the studies. Several studies offer significant relationships and benefits compared to peritoneal dialysis with conventional hemodialysis regarding the Burden of the Disease in the younger population since they have fewer comorbidities [20].

Quality of life (QoL) is a generic concept that reflects the concern for the modification and improvement of attributes due to physical, political, moral, and social environments, as well as health and disease [25]. This concept was created by the World Health Organization in 1948 and reflects physical, mental, and social well-being, not just the absence of disease. In addition, it is recommended to use it for the evaluation of the benefits of therapeutic treatment in patients for more than 20 years [26]. The Karnosfky index can be a complementary scale for patients in RRT, despite being used mainly in oncology, since it helps to rate a person’s ability to perform usual activities, evaluate a patient’s progress after a therapeutic procedure, and determine a patient’s suitability for therapy [27].

However, health-related quality of life (HRQoL) includes the functional effect of a disease on a patient as perceived in the areas of physical, social, and mental well-being [28]. Dialysis has a similar clinical perception component in patients, as they may present with symptoms such as fatigue, tiredness, nausea, pruritus, weight and appetite loss, and even depression and anxiety problems [29]. The main dimensions of the implementation of this project have been to analyze the quality of life through a specific questionnaire to know the different dimensions of the patient with chronic kidney disease (CKD): health, Symptomatology, Effects of the Disease, and Burden of the Disease through a specific questionnaire such as KDQOL-SF, to know the main variables and corresponding analysis [30]. However, we have been able to observe in another study the classification of complex chronic patients through access to the Canadian health system, classifying them according to the type of medical specialization attended, where nephrologists play a fundamental role in determining comorbidities, followed by medical specialists in infectious diseases [31].

It is necessary to unite criteria at a global level in the world to determine the quality of life of patients, as well as the main risk factors and complexity of their conditions, by specialized medical personnel such as nephrologists. The government of Spain in 2020 affirmed that 49.9 million donors were available to perform organ explants, of which the National Transplant Organization (CNT) documented as of 31 December 2019 that 3423 kidney, 1227 liver, 300 cardiac, 419 pulmonary, 76 pancreatic, and 4 intestinal transplants were performed [8]. Despite the research work and activity in the surgical units, the reality is that a significant number of patients persist on the waiting list, pending an organ, in addition to commenting on how the SARS-CoV-2 pandemic has affected this time [9]. Last year, in 2022 the CNT presented the results of the transplants carried out in the historical year of 2020, where 3402 kidney transplants were performed (15% more than the previous year), increasing lung transplants by 15%, pancreas by 12%, cardiac by 3%, and intestinal by 4%. Living-donor kidney transplantation activity this year represented 10% of all transplants [10]. Our study has only evaluated in a single instance, through a question, if there was a failed kidney transplant before, of which 20% of the subjects answered that they had received a failed transplant, and 80% had not received a transplant and were on active dialysis treatment. 80% of patients, taking into account the age profile and associated pathologies such as hypertension, hypercholesterolemia, and diabetes, may have difficult access to the option of kidney transplantation, in addition to the age factor that is decisive since it limits the survival of the donated graft [13].

The results of the present study establish a comparison of the quality of life in patients on hemodialysis and peritoneal dialysis; it is true that the quality of life of transplant patients increases as long as the person is more independent for activities of daily living, in addition to not undergoing dialysis treatment in any of the modalities [14]. But at the same time, we must take into account the work of nursing professionals, both in terms of patient education and psychological care, to improve the process before kidney transplantation [15].

Limitations of the study included the lack of clinical data, such as laboratory analysis, ultrafiltrate, and Kt/V, or social data that could interfere with patient quality of life.

Likewise, the cross-sectional nature of our study does not allow us to draw conclusions about causal or temporal relationships between dialysis modality, the presence of symptoms or symptom improvement, and Quality of Life. Nor is it possible to know if there could be variables prior to the start of therapy that influence the quality of life and condition of the treatment applied. We think there is a need for further studies based on interventions to improve symptoms that could affect quality of life. Longitudinal follow-up studies are also necessary to evaluate quality of life before and after the start of the proposed therapy in order to know exactly how it influences the patient’s daily life. We think it is necessary to apply validated instruments and scales such as the KDQOL-SF and POS-S Renal to know the current situation of the disease as well as the main Symptomatology presented by patients in dialysis units. Patient health education could help to address present and future problems, improve communication with healthcare staff in shared decision-making, and improve quality of life.

## 5. Conclusions

In this study, we found a reference to the quality of life in a dichotomous way, as being good and bad; we can confirm that patients on peritoneal dialysis have a worse quality of life than patients on hemodialysis. With respect to the quality of life of the participants who accessed the study, the presence and management of the symptoms derived from the disease itself—anguish, anxiety, and depression—predominated. It should also be noted that having received a non-failed kidney transplant directly influences the patient’s mood, perception of the disease, and health when compared to previous treatment situations.

## Figures and Tables

**Figure 1 jcm-12-02250-f001:**
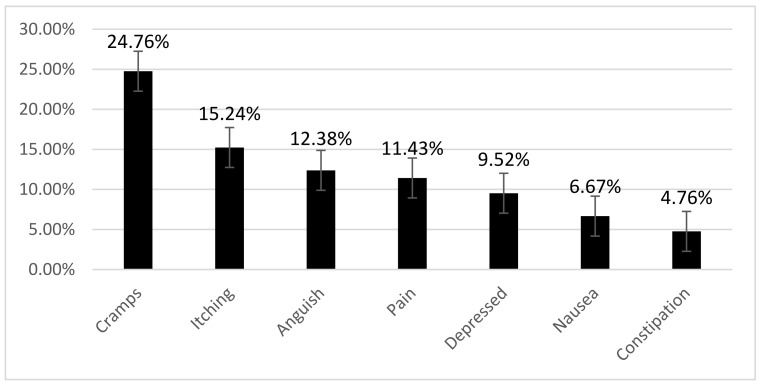
Symptoms that have improved the most in the last week.

**Table 1 jcm-12-02250-t001:** Analytical parameters using the Mann-Whitney U Test.

	Peritoneal Dialysis		Hemodialysis		Total		Size Effect	*t*-Student	*p* Value
	Mean	SD	Mean	SD	Mean	SD			
Dose Kt/V:	.	.	22.37	42.21	22.37	42.21	.	.	.
Hematocrit (%)	31.37	2.88	35.41	6.38	33.80	5.62	5.28	−3.806	0.000
Hemoglobin (g/dL)	12.35	1.72	11.67	1.54	11.95	1.64	1.61	2.119	0.037
Total proteins (g/dL)	5.54	1.34	6.07	0.96	5.85	1.16	1.13	−2.331	0.022
Albumin	3.90	1.03	3.80	0.40	3.84	0.72	0.73	0.591	0.557

SD: Standard deviation; Cohen’s D was used to measure effect size.

**Table 2 jcm-12-02250-t002:** Symptomatology by type of renal replacement therapy.

	Peritoneal Dialysis		Hemodialysis		Total		X^2^	*p*
	n	%	n	%	n	%		
Fever	36	83.72%	4	6.45%	40	38.10%	64.28	<0.001 *
Asthenia	25	58.14%	58	93.55%	83	79.05%	19.22	<0.001
Pyuria	35	81.40%	2	3.23%	37	35.24%	67.99	<0.001 *
Adenopathy	26	60.47%	2	3.23%	28	26.67%	42.54	<0.001 *
Pleural effusion	6	14.63%	4	6.45%	10	9.71%	1.89	0.183 *
WeightLoss	10	23.26%	43	69.35%	53	50.48%	21.59	<0.001
Haematuria	28	65.12%	2	3.23%	30	28.57%	47.66	<0.001 *
Pleuriticpain	9	20.93%	11	17.74%	20	19.05%	0.17	0.68
Headache	31	72.09%	59	95.16%	90	85.71%	11.03	<0.001

* Fisher’s exact test was used.

**Table 3 jcm-12-02250-t003:** Palliative Care Outcome Scale-Symptoms Scale Renal (POS-S Renal) Score.

	None		Slightly		Moderately		Strong		Unbearable		Likert Score	
	n	%	n	%	n	%	n	%	n	%	Media	dt
Sleeping problems	6	5.83	32	31.07	36	34.95	22	21.36	7	6.80	2.92	1.016
Weakness or lack of energy	4	3.81	25	23.81	56	53.33	19	18.10	1	0.95	2.89	0.776
Cramps	4	3.85	44	42.31	29	27.88	24	23.08	3	2.88	2.79	0.942
Feelingdepressed	6	5.77	39	37.50	36	34.62	23	22.12	0	0.00	2.73	0.873
Pain	10	9.52	37	35.24	45	42.86	13	12.38	0	0.00	2.58	0.830
Itching	16	15.38	48	46.15	11	10.58	22	21.15	7	6.73	2.58	1.180
Feeling of distress	10	9.52	51	48.57	24	22.86	19	18.10	1	0.95	2.52	0.931
Skin changes	19	18.10	45	42.86	14	13.33	26	24.76	1	0.95	2.48	1.084
Drowsiness	10	9.62	48	46.15	36	34.62	8	7.69	2	1.92	2.46	0.847
Poor mobility	23	21.90	41	39.05	21	20.00	20	19.05	0	0.00	2.36	1.030
Nausea	24	23.30	43	41.75	23	22.33	12	11.65	1	0.97	2.25	0.977
Restlesslegs	41	39.42	25	24.04	14	13.46	22	21.15	2	1.92	2.22	1.230
Constipation	27	25.71	45	42.86	16	15.24	17	16.19	0	0.00	2.22	1.009
Mouthproblems	40	38.46	31	29.81	9	8.65	23	22.12	1	0.96	2.17	1.194
Diarrhoea	38	36.19	33	31.43	15	14.29	19	18.10	0	0.00	2.14	1.104
Difficultybreathing	24	22.86	58	55.24	16	15.24	7	6.67	0	0.00	2.06	0.807
Loss of hunger	29	27.62	57	54.29	10	9.52	7	6.67	2	1.90	2.01	0.904

**Table 4 jcm-12-02250-t004:** Quality of life according to renal replacement therapy using the Mann-Whitney U Test.

	Peritoneal Dialysis		Hemodialysis		Total		Effect Size *	*t*-Student	*p* Value	Difference of Means	CI (95%)
	Media	dt	Media	dt	Media	dt					
Symptoms or Problem List	49.70	13.14	68.73	11.00	60.94	15.13	11.92	−8.041	0.000	−19.02066	±4.69
Effect of Kidney Disease	34.57	17.89	62.70	13.26	51.18	20.63	15.32	−8.773	0.000	−28.12935	±6.39
Burden of Kidney Disease	31.54	18.60	37.80	12.62	35.24	15.58	15.34	−1.922	0.059	−6.26172	±6.50
SF-12 Physical Component	36.37	5.32	34.25	6.77	35.12	6.27	6.21	1.783	0.078	2.11701	±2.35
SF-12 Mental Component	38.46	9.27	44.30	7.54	41.88	8.75	8.30	−3.532	0.001	−5.83633	±3.27
Total Score	38.13	9.12	49.66	6.97	44.89	9.73	7.93	−7.302	0.000	−11.52683	±3.13

* Cohen’s D was used to measure effect size. CI: Confidence interval 95%.

**Table 5 jcm-12-02250-t005:** Scores in the domains of the KDQOL-SF Scale, according to clinical improvement of symptoms.

	Best Symptoms	n	Symptoms or Problem		List Burden of Illness		Effect of Illness SF-12		Physical Component SF-12		SF-12 Mental Component		Total Score	
			Media	dt	Media	dt	Media	dt	Media	dt	Media	dt	Media	dt
Cramp	No	79	59.34	15.70	34.65	17.15	48.37	21.31 *	35.64	5.83	41.42	9.21	43.88	10.27
	Yes	26	65.79	12.31	37.02	9.34	59.74	15.87	33.48	7.40	43.34	7.06	48.06	7.09
Depressed	No	95	59.91	15.39	34.74	15.78	50.45	20.85	35.05	6.19	41.62	8.92	44.37	9.95
	Yes	10	70.68	7.58 *	40.00	13.24	58.13	17.87	35.77	7.32	44.31	6.75	49.78	5.59
Pain	No	93	60.00	14.97	35.08	15.77	51.00	20.78	35.04	6.32	41.88	8.88	44.61	9.79
	Yes	12	68.23	15.05	36.46	14.56	52.60	20.25	35.79	6.13	41.93	8.00	47.00	9.39
Distress	No	92	59.74	15.19	35.26	15.44	48.77	20.74	35.36	6.41	41.45	8.68	44.13	9.71
	Yes	13	69.39	12.13 *	35.10	17.22	68.27	8.05 *	33.44	5.15	44.90	8.94	50.22	8.44 *
Itching	No	89	62.99	14.93 *	36.03	14.90	55.22	18.35	34.93	6.60	42.73	8.48	46.42	9.05
	Yes	16	49.49	10.77	30.86	18.88	28.71	18.41	36.17	4.00	37.20	8.96	36.49	9.33
Vomiting	No	98	61.19	15.43	35.33	15.98	51.39	20.55	35.32	6.30	41.96	8.78	45.06	9.85
	Yes	7	57.44	10.34	33.93	8.73	48.21	23.17	32.35	5.49	40.87	8.87	42.56	8.14
Constipated	No	100	61.73	14.67*	35.50	15.84	51.49	20.77	35.15	6.37	41.78	8.56	45.15	9.65
	Yes	5	45.00	17.09	30.00	8.15	45.00	18.43	34.58	4.37	43.91	13.05	39.70	11.02

* Significant differences (*p* < 0.05).

**Table 6 jcm-12-02250-t006:** Multivariate analyses.

		No Standardized Coefficients		Standardized Coefficients	*t*	*p*-Value	95.0% Confidence Interval for B	
		B	Desv. Error	Beta			Lower Limit	Upper Limit
Effects of kidney disease	(constant)	−18.073	7.109		−2.542	0.013	−32.177	−3.970
	types of therapies	27.157	2.864	0.642	9.481	0.000	21.474	32.839
	Edad	0.414	0.094	0.297	4.385	0.000	0.227	0.601
Total Score	(constant)	25.616	2.621		9.774	0.000	20.416	30.815
	types of therapies	12.020	1.568	0.608	7.666	0.000	8.909	15.130

The time on peritoneal dialysis was removed from the model for the dependent variable “Total Score”.

## Data Availability

The datasets generated and/or analyzed during the current study are not publicly available due to the fact that they are derived from the proprietary database of a large dialysis organization.

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
