# Peer review of "Quality of Life and Related Factors in Patients Undergoing Renal Replacement Therapy at the Hospital General Universitario de Ciudad Real: Cross Sectional Descriptive Observational Study"

_jcm, 2023, doi:10.3390/jcm12062250_

Round 1

Reviewer 1 Report

Dear authors, thank you very much for your work.

Your study describes symptoms and clinical signs inherent to CKD-V HD/PD organismal maladaptation.

The way you are presenting your results is purely descriptive.

Why is your study important for the interested reader within the nephrological community? It is my opinion that the following issues should be discussed and stressed out:

Isn't the quality of life (QoL) of CKD-V patients equivalently associated with the quality of dialysis? For example, in the pivotal study of Culleton et al. JAMA 2007, nocturnal hemodialysis resulted in improved left ventricular mass function and associated QoL. Which markers addressing dialysis qualtity other than Kt/V did you implement in your study?

Inversely, we as nephrologists, implement therapeutic strategies for patients with many comorbidities (Tonelli et al. JAMA Network Open 2018). When can we tell the difference from QoL symptoms arising as a result of patients´ comorbidities or directly originating from Dialysis quality?

In 2020, the Goverment of Spain, stated that 48.9 donors were available per million population and about 5,500 transplants were performed. How does the relatively high kidney transplantation rate in Spain affects the results presented here? Are the patients presented here renal transplant candidates? Is the QoL for those waiting for renal transplant better compared with those not qualified for transplantation?

Minor spelling and syntax errors should be corrected.

Therefore, I suggest that your work requires a major re-organization.

All of my best regards.

Author Response

Good morning from Spain, 
First of all thank you for your input, as we intend to publish in the magazine. 
We have made changes which are marked in red and we are open to new comments. 
If we need to improve the English of the article if it is accepted we would be willing to do so. 
We look forward to hearing from you. 
Best regards.

Reviewer 2 Report

The introduction is very general. Examples:

“…there are other studies that point out that this improvement depends on the components studied and on clinical and social variables of the patient [11–13].”

“Most of the information available on quality of life in patients with ACKD on RRT comes from other countries, and cannot be extrapolated to our setting due to differences in the health care model, risk profile, relative prevalence of RRT techniques, and even cultural factors.”

Sentences are general, not specific.

The introduction does not contain a number of important articles. Examples:

https://pubmed.ncbi.nlm.nih.gov/15941456/

https://pubmed.ncbi.nlm.nih.gov/8445835/

https://pubmed.ncbi.nlm.nih.gov/18194399/

https://pubmed.ncbi.nlm.nih.gov/21938644/

https://www.ncbi.nlm.nih.gov/pmc/articles/PMC8633691/

These are just examples!

The introduction does not show what the innovation of the research is.

The effect size was not calculated for the statistical tests used. The p-value alone is definitely not enough.

Correlation analysis results are written in different ways. Examples: r= -=.355 p<0.001; r=-0.346 p<0.001.

The statistical test used is not marked in the tables.

In many places, due to e.g. significant inequality of groups, a nonparametric equivalent of a statistical test should be used. The same applies to other assumptions of statistical tests.

The article does not contain more advanced statistical analyses.

The authors did not analyze a number of factors that may affect the obtained results (and they have such a possibility).

Reliability has not been tested.

A discussion is not a discussion. It contains almost no reference to previously published articles. Discussion is not a repetition of the results obtained, but a discussion. Like the introduction, the discussion is written in general terms.

The conclusions are not based on a reliable analysis.

Author Response

(The authors gave the same response as above.)

Round 2

Reviewer 1 Report

Dear authors,

thank you very much for your response and the substantially improved manuscript. Minor text editing is necessary.

All of my best regards.

Author Response

We changed the grammar

Reviewer 2 Report

In many places, due to e.g. significant inequality of groups, a nonparametric equivalent of a statistical test should be used (Mann-Whitney, etc.). Alternatively, other assumptions should be presented for quantitative variables (homogeneity of variance, normality of distribution, specific results of statistical tests must be provided).

Author Response

We have made the changes according to the proposed indications. 
We have made improvements in the grammar to better understand the translation of the language. 
We have performed and extended the statistical tests.
